

# N and P as ultimate and proximate limiting nutrients in the northern Gulf of Mexico: Implications for hypoxia reduction strategies

Katja Fennel[1] and Arnaud Laurent[1]

[1]Department of Oceanography, Dalhousie University, 1355 Oxford Street, Halifax B3H 4R2, Nova Scotia, Canada
*Correspondence to:* Katja Fennel (Katja.Fennel@dal.ca)

**Abstract.** The occurrence of hypoxia in coastal oceans is a growing problem worldwide and clearly linked to anthropogenic nutrient inputs. While the need for reducing anthropogenic nutrient loads is generally accepted, it is costly and thus requires scientifically sound nutrient-reduction strategies. Issues under debate include the relative importance of nitrogen (N) and phosphorus (P), and the magnitude of reduction requirements.

The largest anthropogenically induced hypoxic area in North American coastal waters (of 15,000+/-5,000 km$^2$) forms every summer in the northern Gulf of Mexico where the Mississippi and Atchafalaya Rivers deliver large amounts of freshwater and nutrients to the shelf. A 2001 plan for reducing this hypoxic area by nutrient management in the watershed called for a reduction of N loads. Evidence of P limitation during the time of hypoxia formation has arisen since then, and has opened up the discussion about single versus dual nutrient reduction strategies for this system.

Here we report the first systematic analysis of the effects of single and dual nutrient load reductions from a spatially explicit physical-biogeochemical model for the northern Gulf of Mexico. The model has been shown previously to skillfully represent the processes important for hypoxic formation. Our analysis of an ensemble of simulations with stepwise reductions in N, P and N&P loads provides insight into the effects of both nutrients on primary production and hypoxia, and allows us to estimate what nutrient reductions would be required for single and dual nutrient reduction strategies to reach the hypoxia target. Our

results show that, despite temporary P limitation, N is the ultimate limiting nutrient for primary production in this system. Nevertheless, a reduction in P load would reduce hypoxia because primary production in the region where density stratification is conducive to hypoxia development, but reduction in N load have a bigger effect. Our simulations show that, at present loads, the system is saturated with N, in the sense that the sensitivity of primary production and hypoxia to N load is much lower than it would be at lower N loads. We estimate that reduction of 63% +/- 18% in total N load or 48% +/- 21% in total N&P load are

necessary to reach a hypoxic area of 5,000 km$^2$, which is consistent with previous estimates from statistical regression models and highly simplified mechanistic models.

## 1 Introduction

Coastal eutrophication as a result of anthropogenic nutrient inputs is a growing problem worldwide with negative effects that include hypoxia (Diaz and Rosenberg, 2008), degradation of habitat, and harmful algal blooms (Huisman et al., 2005). The

most important limiting nutrients in aquatic systems are nitrogen (N) and phosphorus (P), and both have major anthropogenic

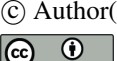


sources (Seitzinger et al., 2010). Fixed N (i.e. N in its bioavailable forms nitrate, nitrite and ammonium) enters aquatic systems mainly through leakage of industrial N-based fertilizer applied in agriculture and through deposition of N resulting from combustion of fossil fuels. P is added to aquatic systems primarily from urban and industrial wastewater as well as fertilizers. Because these nutrients have different sources, the management actions required to reduce one or the other are different. While

P can be controlled by wastewater treatment, control of N requires a decrease in N-based fertilizer use. The need for reducing anthropogenic nutrient inputs to aquatic systems is well recognized, but doing so comes at a significant cost and can be met with substantial political obstacles, in particular with regard to N. Sound nutrient management strategies are thus needed.

There is a long-standing debate about whether controlling only N or P or both of these nutrients is most appropriate for reversing the detrimental effects of eutrophication. As summarized by Conley et al. (2009), in the 1970ies it was established

that P is the primary limiting nutrient in several Canadian lakes (Schindler et al., 2008; Schindler, 1974). Widespread reductions in P loads to North American and European aquatic systems led to improvements in water quality in many lakes, but not in estuarine and coastal systems (see Conley et al., 2009, and references therein). It was concluded that N input needs to be controlled for coastal waters (Howarth and Marino, 2006), and N has been the main target of nutrient load reduction strategies for many estuarine and coastal systems (e.g. Task Force, 2001).

In this context the concept of ultimate versus proximate limiting nutrient is useful. Tyrrell (1999) defines the proximate limiting nutrient as the one that is locally or temporarily limiting primary production; its addition would lead to an immediate enhancement of primary production. In contrast, supply of the ultimate limiting nutrient determines the productivity of a system over long time scales. Clarity about the relevant time scale is important when using these terms. On geological time scales (millennia and longer), P is thought to be the ultimate limiting nutrient of ocean productivity, while N is thought to be

limiting only in the proximate sense (Tyrrell, 1999). On time scales of years to centuries, productivity in the present ocean is clearly limited by the supply of fixed N. In pristine lakes, P is the ultimate limiting nutrient (Schindler et al., 2008). Estuarine and coastal systems that receive heavy nutrient loads can be limited by P or N or both (Conley, 1999; Sylvan et al., 2007). Which nutrient is limiting can vary significantly in time and space (Malone et al., 1996; Sylvan et al., 2007). Establishing for a given estuarine or coastal system which of the two nutrients is the ultimate limiting one (on time scales of years to decades)

should inform the design of sound nutrient-reduction strategies.

The largest hypoxic area in U.S. coastal waters is located in the northern Gulf of Mexico, where hypoxic conditions affect a region of 15,000 km$^2$ +/- 5,000 km$^2$ every summer (Rabalais et al., 2002; Bianchi et al., 2010). Hypoxia in this system is driven by nutrient and freshwater inputs from the Mississippi/Atchafalaya River System, which stimulate high levels of primary production, subsequent decay of organic matter, and vertical density stratification that prevents ventilation (Rabalais

et al., 2002; Bianchi et al., 2010; Yu et al., 2015b). Interannual variability in the size of the hypoxic region is large and hypoxic conditions are restricted to a relatively thin layer above the bottom (Wiseman et al., 1997; Fennel et al., 2013).

N is generally limiting primary production in the Gulf of Mexico; however, observations (Sylvan et al., 2006, 2007) and models (Laurent et al., 2012; Laurent and Fennel, 2014) have shown that in spring and early summer, during the time when hypoxic conditions are established, P is limiting in the Mississippi River plume. The effect of P limitation on hypoxia in this

system has been debated. Scavia and Donnelly (2007) have speculated, based on evidence from other systems (Conley, 1999;



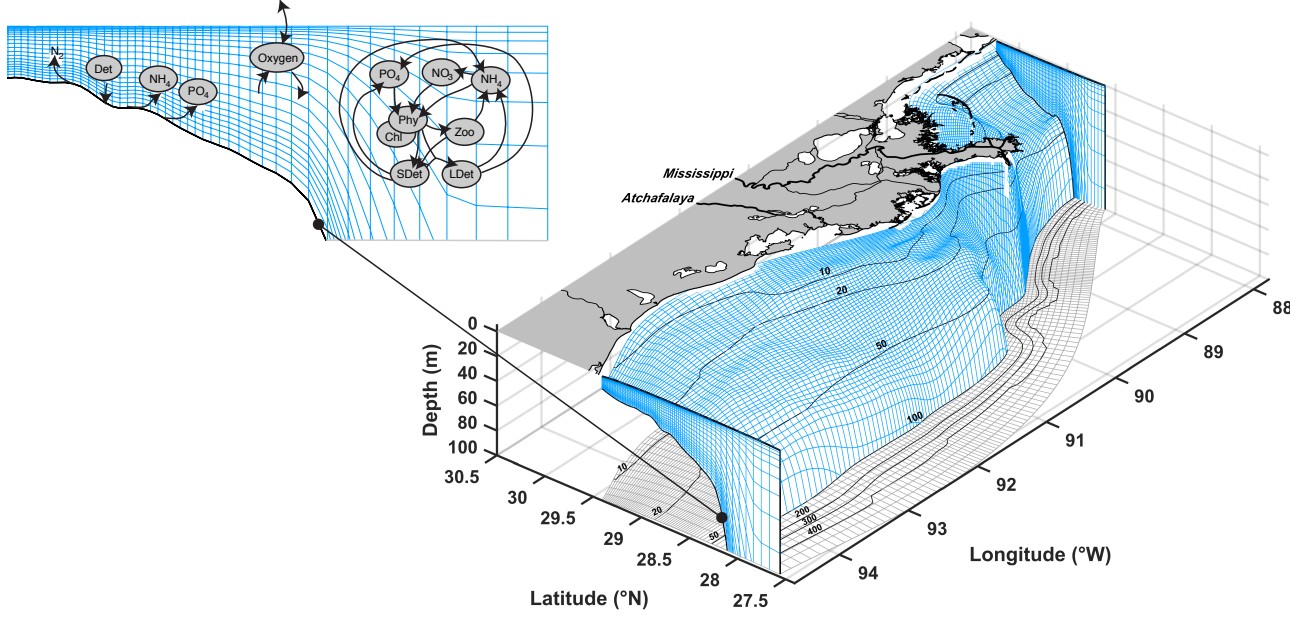

**Figure 1.** ROMS model grid and schematic representation of the biogeochemical variables and processes. The model domain extends to 1000 m depth, but only the top 100 m are shown.

Paerl et al., 2004), that P limitation exacerbates hypoxia by spreading the detrimental effects of elevated N over a larger area. In contrast, the model analysis of Laurent and Fennel (2014) indicates that P limitation mitigates hypoxia in the northern Gulf of Mexico. The model shows that, although P limitation does shift excess N further downstream as suggested by Scavia and Donnelly (2007), the downstream N concentrations are sufficiently diluted that less hypoxia is generated overall.

Despite clear evidence that P is limiting primary production in spring and early summer in the region affected by hypoxia, nutrient reduction strategies for the northern Gulf have long focused on N, implicitly assuming that N is the ultimate limiting nutrient. In 2001, the Action Plan put forth by the Mississippi River/Gulf of Mexico Watershed Nutrient Task Force, an alliance of multiple state and federal agencies and tribes, called for a reduction of the size of the hypoxic zone to a running 5-year mean of 5,000 km$^2$ by the year 2015 through nutrient management in the watershed (Task Force, 2001). The 2001 Action Plan

stated that *"the best current science indicates [..] a 30% reduction [..] in nitrogen discharges [..] is consistent with this goal"* (page 21). No noticeable nutrient load reductions have occurred since then, and the region affected by seasonal hypoxia has not decreased. In the updated Action Plan from 2008, a dual nutrient strategy is called for (Task Force, 2008, page 29). In 2013, the Hypoxia Task Force released a reassessment that called for a *"decrease [in the] scientific uncertainty of nitrogen and phosphorus effects on hypoxia"* (Task Force, 2013, page 49)

Here we use a biogeochemical model for the hypoxic region in the northern Gulf of Mexico (Fennel et al., 2011; Laurent et al., 2012; Laurent and Fennel, 2014; Laurent et al., 2017) to analyze how reduction in N and P loads affect the system. The





motivation for this study is twofold. On the one hand, we aim to determine whether N or P is the ultimate limiting nutrient in this system, and to elucidate how their interplay affects hypoxia development. On the other hand, we address the more practical question of how far N or P loads would have to be reduced to reach the desired reductions in hypoxia. Thus far, the primary modeling tool for defining nutrient reduction targets has been statistical modeling that relates spring nutrient loads to summer

hypoxic extent (Scavia et al., 2003; Greene et al., 2009; Forrest et al., 2011; Turner et al., 2012; Obenour et al., 2015). These models are not spatially explicit, and ignore or highly simplify the mechanisms underlying hypoxia generation. Variations in spring nutrient load, although significantly correlated with summer hypoxic area, explain only 24% of variability in hypoxic area in the study of Forrest et al. (2011). When other factors like directional wind strength and freshwater discharge are incorporated as independent variables the correlation improves markedly (Forrest et al., 2011). This illustrates the importance

of variations in atmospheric forcing and circulation patterns in determining hypoxic conditions on the shelf, and suggests that a spatially explicit, mechanistic approach is valuable. While our study is specific to the northern Gulf of Mexico, the findings should also be relevant to other coastal shelf systems that receive high nutrient loads, e.g. the North Sea and the East China Sea.

## 2   Methods

Our biogeochemical model (Figure 1) uses a high-resolution implementation of the Regional Ocean Modeling System (ROMS; Haidvogel et al., 2008) for the northern Gulf of Mexico coupled with the relatively simple N cycle model of Fennel et al. (2006). The original N cycle model has been expanded to include phosphate as additional nutrient (Laurent et al., 2012), dissolved oxygen (Fennel et al., 2013) and river-derived dissolved organic matter (Yu et al., 2015b). An up-to-date description of the model equations is provided in the supplemental information of Laurent et al. (2017). The model is configured for the

shelf region of the northern Gulf of Mexico that frequently experiences hypoxia (Figure 1). It has been extensively validated by comparing standing stocks and rates against available measurements and has been shown to represent the biogeochemical dynamics of the system well (see Fennel et al., 2011, 2013; Laurent et al., 2012; Laurent and Fennel, 2014; Yu et al., 2015b).

The model is forced with 3-hourly winds from the NCEP North American Regional Reanalysis (Mesinger et al., 2006), climatological surface heat and freshwater fluxes from da Silva et al. (1994), and daily freshwater discharge from the Missis-

sippi and Atchafalaya Rivers recorded by the U.S. Army Corps of Engineers at Tarbert Landing and Simmesport, respectively. Inputs of nutrients, and particulate and dissolved organic matter are based on monthly flux estimates from the U.S. Geological Survey (Aulenbach et al., 2007). Model simulations start on 1 January 2000 and end on 31 December 2016.

In our previous studies, we have used several, qualitatively different parameterizations for the interaction between sediment and overlying water column: an instant remineralization (IR) parameterization, which assumes that all organic matter is rem-

ineralized immediately upon reaching the sediment (Fennel et al., 2013), an empirical parameterization where sediment oxygen consumption and nutrient efflux from the sediment depend on bottom water temperature and oxygen (e.g., Yu et al., 2015a), and sediment flux parameterizations based on a metamodel analysis of a diagenetic model (Laurent et al., 2016). The simple empirical parameterization, in particular, has proven useful as a computationally efficient and accurate bottom boundary layer




(Fennel et al., 2013, 2016). However, neither the metamodel nor the empirical parameterization are appropriate for this study because they do not explicitly consider the depositional flux of organic matter to the sediment.

Since the objective here is to quantify the response of hypoxia to nutrient reductions, the IR parameterization has to be used. One known issue with IR is that the simulated hypoxic area is biased low compared to the empirical parameterization and observations. In order to address this problem, we debiased the simulated hypoxic area by normalization against the observation-based estimates of Obenour et al. (2013). We calculated the average ratio between observed and simulated hypoxic area during the hypoxic monitoring cruises (Rabalais et al., 2002) for the 12 years for which the simulation period and data set overlap and applied this ratio to all simulated hypoxic area estimates.

We conducted one 17-year simulation using the current nutrient loads as described above. Then we repeated the same simulation with reduced loads of total nitrogen (TN), dissolved inorganic phosphorus (DIP) and reduced loads of both (TN&P). In all three cases the loads were decreased by 20%, 40%, 60% and 80%. This resulted in a total of 13 simulations. The freshwater discharge was not changed.

## 3   Results

### 3.1   Seasonal evolution of nutrients, primary production and hypoxia area

To illustrate the effects of nutrient reductions, we first compare time series of shelf-averaged surface nutrients, primary production and bottom-water hypoxia in 2009. These are shown in Figure 2 for four simulations: the one with current nutrient loads, the one with a 60%-reduction in TN load, the one with a 60%-reduction in DIP load, and the one with a 60%-reduction in both.

In the simulation with current nutrient loads (dark orange lines in Fig. 2), the shelf-averaged surface nitrate concentration is high in winter with ∼10 mmol N/m$^3$. Nitrate increases further in spring and early summer to 18 mmol N/m$^3$ due to nutrient input in spring, then decreases rapidly in mid summer to about 5 mmol N/m$^3$ as a result of algal uptake and continues to decrease more slowly in late summer and early fall until reaching its minimum in October and beginning to increase again in late fall.

Phosphate behaves differently, reaching its maximum concentration between 0.6 and 0.8 mmol P/m$^3$ in winter and its minimum concentration of 0.3 mmol P/m$^3$ in summer. The amplitude of seasonal changes (i.e., the ratio between maximum and minimum concentration) is smaller for phosphate than nitrate. The difference in the seasonal cycles of nitrate and phosphate is a result of P limitation, which, according to observations (Sylvan et al., 2006) and model simulations (Laurent et al., 2012), occurs in plume waters during spring and early summer after the annual maximum in riverine nutrient input. P limitation of primary production results in an accumulation of inorganic nitrogen in early summer. In late summer and early fall, as river-derived high-nitrate waters mix with marine waters that hold an excess of phosphate relative to nitrate, P limitation is relieved and most of the accumulated nitrate is eventually taken up until minimum concentrations are reached in fall.

In the simulation with 60%-reduction in DIP load (light orange lines in Fig. 2), average phosphate is, as expected, much smaller than for current loads. This makes P limitation in the plume in spring and early summer more severe and further ampli-





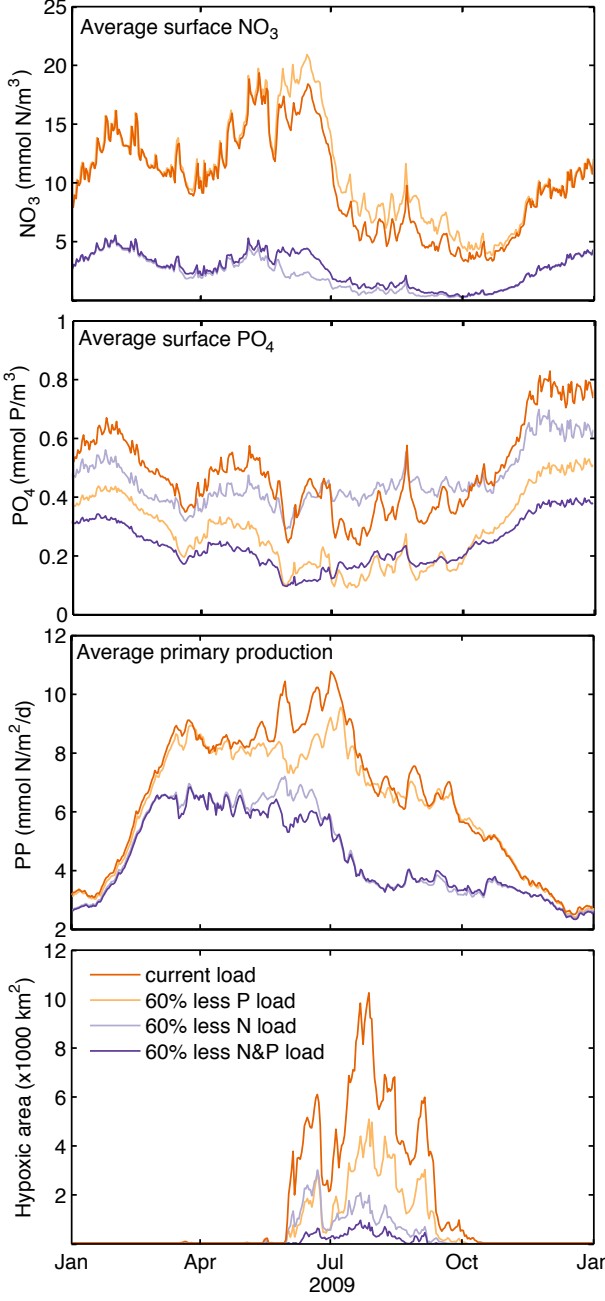

**Figure 2.** Evolution of shelf-averaged surface nitrate and phosphate concentrations, primary production and bottom-water hypoxic area for 2009 from the simulation with current loads (dark orange), 60% reduction in TN load (light orange), 60% reduction in DIP load (light purple), and 60% in both (dark purple).





fies the accumulation of surface nitrate in summer (i.e., average nitrate concentrations in summer are larger in the simulation with P reductions than in the simulation without nutrient reductions).

A 60%-reduction in TN load (light purple lines in Fig. 2) results in the smallest surface nitrate concentrations of all four simulations. The June nitrate maximum of 18 mmol N/m$^3$ in the current-load simulation is reduced to 2.5 mmol N/m$^3$, a decrease

of 86% that is much larger than the 60% reduction in river TN load, i.e. nitrate accumulation in summer is less pronounced. In the simulation where both TN&P loads are reduced by 60% (dark purple lines), the summer nitrate concentrations are slightly larger than in the case where only TN load is reduced, indicating again that more severe P-limitation in the summer results in nitrate accumulation.

Shelf-averaged primary production is, as expected, largest in the current-load simulation. Compared to this, primary produc-

tion slightly decreases in early summer when the DIP load is reduced, but decreases significantly from March to November for a reduced TN load. In the simulation with TN&P load reduction, primary production is similar to the result for the TN load reduction, except for a brief period in early summer when primary production is even lower. The effect of DIP load reductions on primary production is thus minor as illustrated by the two pairs of simulations (current load versus DIP load reduction, and TN load reduction versus TN&P load reduction), while the TN load reduction has a big effect. Annually integrated values of

primary production for 2009, i.e. the year shown in Figure 2, and averaged over the years 2000 to 2016 are listed in Table 1.

Hypoxic conditions occur in all four simulations shown in Figure 2 from early June to the end of September, but the simulated spatial extent of hypoxia is different in all of them. Hypoxia is most expansive in the current-load simulation, decreases significantly in the simulation with P load reduction, is reduced further in the simulation with N load reduction, and is smallest in the simulation with N&P load reduction. The values of annually integrated hypoxic area, $H$, are given in Table 1.

Given the relatively minor effect of DIP load reductions on shelf-averaged primary production, the large sensitivity of the simulated hypoxic area to DIP load is perhaps surprising. However, as discussed in previous publications, hypoxic conditions are spatially and temporally constrained by the "stratification envelope," i.e. the existence of a stratification regime that is conducive to hypoxia by preventing ventilation of bottom waters (Hetland and DiMarco, 2008). Hypoxic extent is thus sensitive to the spatio-temporal alignment between peak primary production and the occurrence of the stratification envelope (Laurent

and Fennel, 2014). Temporal and spatial shifts, and changes in magnitude of peak primary production in summer, which result from variations in P load, can thus have a notable effect on hypoxia without altering shelf-averaged primary production significantly.

Next, we systematically compare annually integrated values of primary production and hypoxic area for the different nutrient load reductions and determine their sensitivity to nutrient load decreases.

**3.2  Sensitivity of primary production and hypoxia to nutrient load reductions**

We define the dimensionless sensitivity $S$ of a system property (e.g. shelf-averaged PP or $H$) to nutrient load reduction as the ratio between the change in this property (in %) to the imposed change in nutrient load (in %). In other words, a sensitivity of PP equal to 1 implies that for a 10% decrease in nutrient load, a 10% decrease in PP can be expected. If the sensitivity is less





than 1, a 10% decrease in nutrient load would bring about a decrease in PP of $S \times 10\%$, i.e. smaller than 10%. When $S$ is larger than 1, the change in PP would be larger than 10%.

Sensitivities of shelf-averaged PP and $H$ over May N load are shown in Figure 3. At high loads of around $10 \times 10^9$ mol N, the sensitivity of PP to TN load reduction is relatively small around 0.4. When loads are reduced to less than $8 \times 10^9$ mol N or

less than $5 \times 10^9$ mol N the sensitivity increases to 0.65 or almost 1, respectively. Shelf-averaged PP is relatively insensitive to reductions in DIP load.

At current DIP loads, the sensitivity of $H$ to TN loads reductions is larger than the sensitivity of PP, and also increases for smaller nutrient loads from 1.1 at loads around $10 \times 10^9$ mol N to 1.6 for loads less than $5 \times 10^9$ mol N. At current loads, the sensitivity to DIP load reduction is 0.78, smaller than the sensitivity to TN load reduction. In other words, a 10% decrease in

TN load would shrink the hypoxic area more than a 10% decrease in DIP load. At lower DIP loads, the sensitivity of $H$ to N load reduction decreases.

These results indicate that nutrient load reductions would likely bring about larger decreases in hypoxic area than in PP. The system is currently approaching N saturation, i.e. larger nutrient loads would not increase PP significantly, because the system is already saturated in N. This also implies that initial nutrient reductions from present loads will have a more modest effect

than similar reductions would accomplish at lower N loads.

For the year 2009, which we have considered thus far, the response to nutrient load reductions is well behaved and suggests that predictive relationships can be derived. However, it is important to recognize that interannual variability in the phenology of freshwater and nutrient discharges and in shelf circulation results in very different hypoxia expressions from year to year (e.g., Feng et al., 2014). Next we account for interannual variability and derive predictive relationships that consider this source

of uncertainty.

### 3.3    Defining nutrient reduction targets

In Figure 4 we show the simulated hypoxic area in summer in comparison with the corresponding observed estimates of Obenour et al. (2013) for the 12 years that both records overlap. This comparison illustrates that the model roughly agrees with the observations and has a similar response to variations in May N load.

May N load varies considerably from year to year (Figure 4), but even when comparing years with similar load, hypoxic area estimates are highly variable in the observations and the model. A large degree of interannual variability, even when nutrient loads are similar, is not surprising given the oceanographic characteristics of the system. The region that is prone to hypoxia is an open shelf system influenced by a highly dynamic river plume. The evolution of the plume distribution, vertical stratification and hypoxia are strongly affected by shelf circulation, which is determined by variable surface forcing (e.g., Feng et al., 2014),

the passage of atmospheric disturbances with high wind, and meso- and submesoscale dynamics (e.g., Marta-Almeida et al., 2013; Mattern et al., 2013).

We account for the uncertainty resulting from this interannual variability by using all 17 years of our simulations. Summer hypoxic area for all years and all simulations with TN load reductions (but without reductions in DIP load) are shown in Figure 5 (top). The orange squares with error bars are means +/- one standard deviation of the binned data. At high nutrient loads of





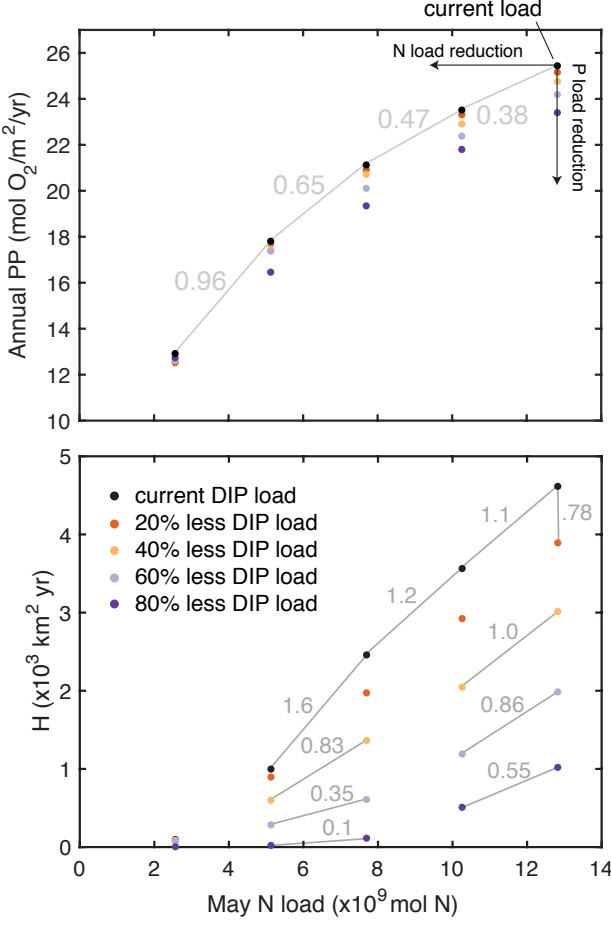

**Figure 3.** Shelf-averaged annual primary production (PP) and annually integrated hypoxic area $H$ for 2009 plotted over May N loads for 20%, 40%, 60% and 80% reductions in TN and DIP loads. Gray lines and numbers indicate sensitivity to load reductions ($S$) as defined in the text.

$>10\times10^9$ mol N, standard deviations are large and hypoxic area is relatively insensitive to N load. In other words, the system is saturated in N. Below loads of $10\times10^9$ mol N, hypoxic area decreases with N load reductions.

Using piece-wise linear regression to estimate at which TN load the hypoxic area would be reduced to 5,000 km$^2$ yields an estimate of 4.3 +/- 2.1$\times10^9$ mol N (a reduction of 63.2% +/- 17.9% of the current TN load). The same analysis is repeated for the simulations with proportional reductions in TN&P (Figure 5; bottom). In this case, the targeted hypoxic area of 5,000 km$^2$ would be reached at a load of 6.1 +/- 2.5$\times10^9$ mol N (a reduction of 48.4% +/- 21.1% of the current TN&P load).



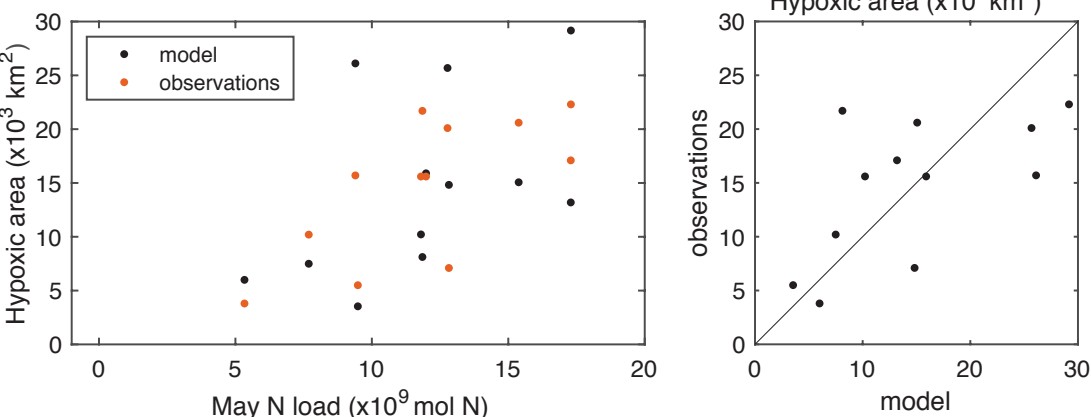

**Figure 4.** Left) Simulated mid-summer hypoxic area for current nutrient loads plotted over May N load in comparison to observed values by Obenour et al. (2013) for the years 2000 to 2011. Right) Observed over simulated mid-summer hypoxic area for the same years.

### 3.4 Nutrient targets in comparison to previous studies

We now compare our estimates of the reductions in nutrient load that are necessary to reach a summer hypoxic area of 5,000 km$^2$ with previous estimates from the literature (see Table 2). The first published estimate was provided by the Hypoxia Taskforce in their first Action Plan (Task Force, 2001), which states that according to the then available science a 30%

reduction should be sufficient. The document does not elaborate on how this estimate was made. Since then a number of estimates have been published using simple mechanistic models and purely empirical regressions.

Scavia et al. (2003, 2013) developed a one-dimensional model that simulates oxygen downstream of organic matter sources, and accounts for oxygen consumption due to organic matter decomposition and resupply by ventilation. The model of Obenour et al. (2015) is a mass-balance model that simulates nutrient-stimulated primary production, organic matter sedimentation,

decomposition of organic matter in water column and sediments, and ventilation. Both models are based on mechanistic assumptions, but highly simplify the physical and biogeochemical processes affecting oxygen. The models of Greene et al. (2009), Forrest et al. (2011) and Turner et al. (2012) are purely empirical, the first two based on multivariate linear regressions and different combinations of predictive variables, the latter based on a curvilinear fit between nutrient load and hypoxic area.

The comparison in Table 2 shows that the estimates of necessary reductions have increased over time, the initial Taskforce

estimate being the lowest. This increase does not only apply to the aggregate of estimates; it is also evident where updated estimates of individual models were published over time (see, e.g., Scavia's 2003 and 2013, and Forrest's 2011 and 2017 estimates). The increase may in part be due to a refinement of models and in part due to the growth of the available data set.

For some combinations of predictor variables, the multiple linear regressions of Greene et al. (2009) and Forrest et al. (2011) predict that even a 100%-reduction in nutrient loads would be insufficient to reach 5,000 km$^2$ of hypoxic area, which seems



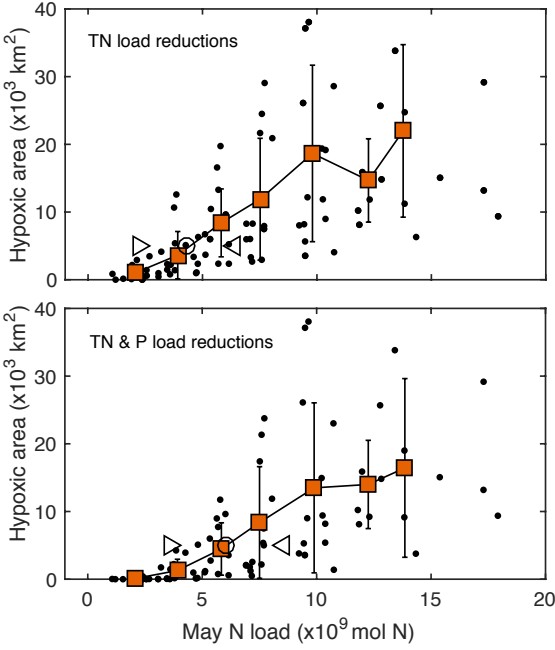

**Figure 5.** Simulated mid-summer hypoxic area plotted over May N load for nutrient load reduction scenarios. Top) Simulated hypoxic area for current loads and all simulations with reduced N loads (black dots), binned means +/- one standard deviation (orange squares with error bars) and value at which target of 5,000 km$^2$ hypoxic area would be reached +/- one standard deviation (open circle and open triangles). Bottom) Same as middle panel but for proportional reductions of TN&P load.

unrealistic. These results illustrate the difficulty in extrapolating outside the historically observed range, which is especially fraught when hypoxia sensitivity to nutrient load changes as our model suggests it does (see section 3.2).

In the recent estimates by Scavia et al. (2017), three of the four different models (UM, NCSU and LSU) are remarkably consistent; however, all assume reductions to different pools of nitrogen load.

5      Our model's estimates are consistent with previous estimates in several respects. Our TN reduction estimate is very close to those of Scavia et al. (2013, 2017). Our model suggests that a proportional reduction of P loads would reduce the necessary load reduction by 15% (from 63% to 48%) about twice the 8% reduction (from 50% to 42%) that Greene et al. (2009) predicted with their model 11.

## 4    Conclusions

10    This is the first analysis for the northern Gulf of Mexico that uses a spatially explicit physical-biogeochemical model to assess the effects of nutrient load reductions and estimate the load reduction targets to reach the hypoxia reduction goal set by the Task Force (2001). An ensemble of biogeochemical scenario simulations for the hypoxic zone in the northern Gulf of Mexico,





where riverine loads of TN, DIP or both were reduced in a stepwise manner, shows that system-wide primary production is much more sensitive to variations in N load than P load. This is consistent with the notion that N is the ultimate limiting nutrient in this system, while P is limiting only in a proximate sense. The sensitivity of primary production to TN load varies. At the high end of the range of current loads the sensitivity is relatively low ($\sim$0.4), but increases to almost 1 when TN load is

reduced by at least 60%. This indicates that the system is essentially saturated in N.

Although P-load reductions have little effect on overall primary production, they lead to a significant decrease in hypoxia. This is because intensified P-limitation in summer decreases the peak in production of organic matter, thus reducing the supply of organic matter in the shelf region where density stratification is conducive to hypoxia. However, hypoxia is more sensitive to N-load reductions than reductions in P. As with primary production, the sensitivity of hypoxia to N load reduction changes for

different N loads. Consequently, statistical extrapolation outside the historically observed range of conditions should be treated with caution.

Previously published, simple predictive models relate summer hypoxic area to May N load (see Table 2), but interannual variability in hypoxic area is large, even among years with similar May N load, because of year-to-year variations in ocean circulation and in the phenology of river inputs. By considering an ensemble of 17-year simulations we account for interan-

nual variability when estimating the load reductions that would be required to reach the goal of 5,000 km$^2$. Piecewise linear regression of simulated summer hypoxic area against May N load suggests that load reduction of 63% +/- 18% of TN load or 48% +/- 21% TN&P load are necessary. In other words, a dual nutrient strategy would be most effective in reducing hypoxia. These estimates are consistent with the previously published estimates.

*Competing interests.* The authors have no competing interests.

*Acknowledgements.* This work was supported by the NOAA Coastal and Ocean Modeling Testbed (COMT). NGOMEX publication no. XXX.



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





**Table 1.** May total nitrogen (TN) load, simulated annual primary production (PP) and annually integrated hypoxic area $H$ for 2009 (normal font) and averaged over the years 2000 to 2016 (in bold) for current loads and selected nutrient load reduction simulations.

| | current load |
|---|---|
| May TN load ($10^9$ mol N) | 12.8 (year 2009) <br> **11.7** (avg 2000-2016) |
| Annual PP (mol $O_2$ /m$^2$ /yr) | 25.4 <br> **26.7** |
| Hypoxic area $H$ ($10^{13}$ km$^2$ yr) | 4.61 <br> **3.83** |

| | DIP load reduction | | | |
|---|---|---|---|---|
| | 20% | 40% | 60% | 80% |
| May TN load ($10^9$ mol N) | 12.8 <br> **11.7** | same <br> same | same <br> same | same <br> same |
| Annual PP (mol $O_2$ /m$^2$ /yr) | 25.1 <br> **26.3** | 24.7 <br> **25.7** | 24.2 <br> **24.9** | 23.4 <br> **24.0** |
| Hypoxic area $H$ ($10^{13}$ km$^2$ yr) | 3.89 <br> **3.10** | 3.01 <br> **2.31** | 1.99 <br> **1.50** | 1.02 <br> **0.77** |

| | TN load reduction | | | |
|---|---|---|---|---|
| | 20% | 40% | 60% | 80% |
| May TN load ($10^9$ mol N) | 10.3 <br> **9.39** | 7.69 <br> **7.04** | 5.12 <br> **4.70** | 2.56 <br> **2.35** |
| Annual PP (mol $O_2$ /m$^2$ /yr) | 23.5 <br> **25.2** | 21.2 <br> **22.9** | 17.8 <br> **19.1** | 12.9 <br> **13.5** |
| Hypoxic area $H$ ($10^{13}$ km$^2$ yr) | 3.56 <br> **2.98** | 2.46 <br> **2.06** | 1.00 <br> **1.01** | 0.10 <br> **0.16** |

| | TN&P load reduction | | | |
|---|---|---|---|---|
| | 20% | 40% | 60% | 80% |
| May TN load ($10^9$ mol N) | 10.3 <br> **9.39** | 7.69 <br> **7.04** | 5.12 <br> **4.70** | 2.56 <br> **2.35** |
| Annual PP (mol $O_2$ /m$^2$ /yr) | 23.2 <br> **24.9** | 20.7 <br> **22.3** | 17.4 <br> **18.6** | 12.7 <br> **13.3** |
| Hypoxic area $H$ ($10^{13}$ km$^2$ yr) | 2.92 <br> **2.35** | 1.36 <br> **1.05** | 0.29 <br> **0.24** | 0.004 <br> **0.006** |





**Table 2.** Previously estimated nutrient load reductions necessary to the reach the hypoxic area target of 5,000 km$^2$ and estimates from this study. In the right column, numbers in square brackets are 95% confidence intervals. Numbers that follow +/- are standard deviations. TN refers to total nitrogen. P refers to phosphate. NO$_x$ refers to nitrate+nitrite. BO refers to bioavailable nitrogen as defined in the referenced study.

| Reference | | Estimated load reduction |
|---|---|---|
| Taskforce (2001) | | 30% N load |
| Scavia et al. (2003)[§] | | 40-45% TN load |
| Scavia and Donnelly (2007)[§] | | 37-45% TN load |
| | | 40-50% P load |
| Greene et al. (2009)[†] | model 11 | 50% NO$_x$ load |
| | | 42% NO$_x$&P load |
| | model 12 | >100% NO$_x$ load |
| | | 42% NO$_x$&P load |
| Forrest et al. (2011)[†] | UEDC | 68% NO$_x$ load |
| | UEN | >100% NO$_x$ load |
| Turner et al. (2012)[‡] | | 57% TN load[#] |
| Scavia et al. (2013)[§] | | 62% TN load |
| Scavia et al. (2017) | | |
| | UM[§,1] | 58% [49-70%] TN load |
| | NCSU[§,2] | 56% [50-62%] BN load |
| | LSU[‡,3] | 56% [50-64%] NO$_x$ load |
| | VIMS[†,4] | 80% +/- 70% NO$_x$ load |
| this study | | 63% +/- 18% TN load |
| | | 48% +/- 21% TN&P load |

§ highly simplified mechanistic model

† empirical multi-linear regression model

‡ curvilinear regression model

1 same model as in Scavia et al. (2013)

2 same model as in Obenour et al. (2015)

3 same model as in Turner et al. (2012)

4 same model as in Forrest et al. (2011)

# converted from 70,000 metric tons