# Peer review of "N and P as ultimate and proximate limiting nutrients in the northern Gulf of Mexico: Implications for hypoxia reduction strategies"

_Biogeosciences, 2017_

## Referee Comment (RC1) · Anonymous Referee #1 · 7 Mar 2018

General Comments: This is an excellent and timely manuscript by Fennel and Laurent on an important problem. It is well written and presentation quality is high. The authors are active researchers in the Gulf hypoxia community and are well aware of the related work and current efforts to improve scientific understanding of hypoxia in the northern Gulf and reduce nutrient inputs to the system. However, I do not find the concept of ultimate and proximate limiting nutrients to be a particularly compelling framework for this paper (and it is only briefly mentioned in the conclusion). Instead, the significance of the work is the application of spatially-explicit physical-biogeochemical models to elucidate the influence of dual nutrient management strategies on the northern Gulf ecosystem. It is encouraging to see that the magnitude of nutrient reductions predicted

by the model align with prior regression/statistical approaches that have formed the basis of nutrient management efforts in this system.

Specific Comments: 1. Pg 1, line 23. The statement that coastal eutrophication from nutrient inputs is a growing problem.... ignores decades of observations of the problem of eutrophication and nutrient pollution. Perhaps it is now appropriate to state that coastal eutrophication from nutrient inputs is a long standing problem...

2. Pg 2, line 13-14. N is not the main target of nutrient load reductions for Gulf hypoxia. In advance of the 2008 Hypoxia Action, EPA and the Hypoxia Task Force convened a special Hypoxia Advisory Panel thru the EPA SAB to review the science and provide recommendations for reducing Gulf hypoxia (see reference below). The SAB recommended a dual nutrient strategy of reducing N&P loads by 45%. The Gulf Hypoxia Task Force endorsed the dual nutrient reduction strategy and since the 2008 Action Plan a dual N&P load reduction has been and remains the target. The focus on N&P will have local in-stream water quality benefits as well as downstream water quality benefits in the Gulf consistent with our conceptual understanding and model predictions.

U.S.EPA 2007. Hypoxia in the Northern Gulf of Mexico. An update by the EPA Science Advisory Board. EPA-SAB-08-003. Washington D.C.

3. Pg 4, line 26-27. The monthly flux estimates described by Aulenbach et al., 2007 include two different regression model approaches; adjusted maximum likelihood estimates (AMLE) and the composite method. Please indicate which method was used.

4. Pg 5, line10. Please provide the rationale for choosing TN and DIP for load reduction scenarios rather than DIN and DIP, TN and TP or NOx and PO4, or something else. The literature is inconsistent in what is used for load reduction scenarios to predict Gulf hypoxic zone size. In the lower MS River, nitrate/nitrite comprises about 65% of the TN pool, and it is the nitrate/nitrite pool that has increased several-fold due to anthropogenic activities in the MS basin.

5. Pg 10, line 3-6. The 30% N reduction goal described in the 2001 Hypoxia Action Plan comes from the work of V. Bierman and colleagues as part of the Topic 4 report for the first integrated assessment of hypoxia in the northern Gulf (see references below). Bierman et al used a version of the WASP model tailored to the Gulf hypoxia zone (which he called the NECOP model) to evaluate N and P reduction scenarios from 10% - 70%.

CENR (Committee on Environment and Natural Resources) 2000. Integrated assessment of hypoxia in the northern Gulf of Mexico. National Science and Technology Council Committee on Environment and Natural Resources, Washington, D.C., USA.

Brezonik, P.L., Bierman, V.J., Alexander, R., Anderson, J., Barko, J., Dortch, M., Hatch, L., Hitchcock, G.L., Keeney, D., Mulla, D., Smith, V., Walker, C., Whitledge, T., and Wiseman, W.J., 1999, Effects of reducing nutrient loads to surface waters within the Mississippi River Basin and Gulf of Mexico, Topic 4 Report for the Integrated Assessment on Hypoxia in the Gulf of Mexico: Silver Spring, Maryland, National Oceanic and Atmospheric Administration, NOAA Coastal Ocean Program Decision Analysis Series No. 18.

6. Pg 10, line 12. Suggest including reference to the statistical model of Turner et al. 2006 (see reference below), which helped to stimulate a series of additional statistical modeling approaches that you identify in the paper.

Turner, R. E., N. N. Rabalais, and D. Justic. 2006. Predicting summer hypoxia in the northern Gulf of Mexico: riverine N, P, and Si loading. Marine Pollution Bulletin 52:139–148.

7. Pg 10, lines 18-19. The statement that Greene et al 2009 (and Forrest et al 2011) predict >100% nutrient load to achieve the 5000 km2 goal is incorrect. Greene et al 2009 used 50% N and 45% N&P load reduction scenarios and found that model 11 (which used AMLE load estimates) could achieve the hypoxia target, whereas, model 12 (which used composite method load estimates) would not achieve the hypoxia target. These differences were attributed to differences in the AMLE and COMP load estimation methods, which influenced the model parameter coefficients. Thus, it would be correct to state on line 19 (and in Table 2) that >50% N reduction would be needed to achieve the hypoxia target using model 12.

8. Pg 17 Table 2. There is an inconsistency in abbreviations – figure caption state 'BO' refers to bioavailable N where the NCSU model in Scavia et al 2017 shows 'BN'. Finally, Greene et al 2009 used 'TP' not 'P' in their dual nutrient regression models.

---

## Referee Comment (RC2) · Anonymous Referee #2 · 8 Mar 2018

This manuscript reports the first systematic analysis of the effects of single and dual nutrient load reductions from a spatially explicit physical-biogeochemical model for the northern Gulf of Mexico (GoM). Their manuscript is the next important step in their modeling efforts in the GoM. The manuscript tackles an important question regarding nutrient reduction strategies, it is well written and their work reports important simulations regarding dual nutrient reductions. They come to several additional important conclusions about the behavior of the GoM ecosystem including that reductions are more effective at reducing the area of hypoxia than in reducing primary production, and that the Gulf of Mexico system is saturated with nitrogen.

[Figure]

Some comments:

The US has not been able to make real nutrient reductions despite decades of voluntary controls. Since the GoM is N saturated, does that mean that it can't get worse as N loads continue to increase. Are we really at the maximum area given the current conditions and climate?

Pg 1, line 8 "Evidence of P... since then" is awkward.

Pg 3, line 11 One question regarding hypoxia is the legacy of a higher sediment respiratory demand following the build up of organic carbon stores in sediments with eutrophication (Turner et al. 2008) whereby repeated hypoxic events lead to an increased susceptibility of further hypoxia and accelerated eutrophication. I know one group in the GoM that believes that this process can not happen because of the physical conditions on the shelf would prevent the accumulation of organic matter stores. However, there are studies from the Baltic addressing the importance of the legacy of carbon and nutrients in the sediments. Could this be a factor in the GoM?

Pg 4, line 17 Revise to "as an additional"

Pg 7, "Sensitivity" – this is an interesting concept...

Turner and Rabalais have examined the role of Si in influencing diatom growth and hence the sedimentation of organic matter. Have you tried any simulation with dissolved silica?

---

## Referee Comment (RC3) · Anonymous Referee #3 · 14 Mar 2018

1. This is an important and generally well-written modeling paper which moves the discussion of nutrient loading and Gulf hypoxia further along. In addressing the question of ultimate vs. proximate limiting nutrient, which I don't object to, probably the major point to make is that there would be no P limitation on the Louisiana shelf without the excessive loading of N from the Mississippi River, which the authors refer to as "saturating", an appropriate and significant term. 2. Though the 2008 Hypoxia Task Force Action Plan, which is still the current goal though delayed in time, is mentioned on page 3 line 12, its load reduction goals (45% N and P) are not listed in Table 2 nor discussed in the text other than as a dual nutrient strategy, and the 2007 EPA Science Advisory Report, cited by another reviewer, which is the basis for these reduction goals, is not

mentioned or cited. This SAB report was a 300-page major review of the science status at the time and the authors cite some of the papers important to this SAB review, but not the document itself. 3. Unless I am reading it wrong, there is an inconsistency between the written legend to Fig. 2 and the labeled legend in the hypoxic area graph. The label says -60% P is light orange, and the written legend says that light orange is the -60% TN. The labeled legend in the figure appears to be correct and consistent with the following text, while the written legend appears incorrect. This labeling should be confirmed to be consistent throughout or it will prove very confusing. 4. I agree that the use of TN and DIP river load reductions is confusing and probably has more of a historic rather than scientific origin. That said, much of the river Total P load is particulate which then is solubilized when reaching the Gulf.

---

## Author Comment (AC1) · 3 Apr 2018

**Responses to Comments by Reviewer #1**
https://doi.org/10.5194/bg-2017-470-RC1, 2018

Reviewer comments are pasted in their entirety in black font. Responses are in blue font.

**General Comments:** This is an excellent and timely manuscript by Fennel and Laurent on an important problem. It is well written and presentation quality is high. The authors are active researchers in the Gulf hypoxia community and are well aware of the related work and current efforts to improve scientific understanding of hypoxia in the northern Gulf and reduce nutrient inputs to the system. However, I do not find the concept of ultimate and proximate limiting nutrients to be a particularly compelling framework for this paper (and it is only briefly mentioned in the conclusion). Instead, the significance of the work is the application of spatially-explicit physical-biogeochemical models to elucidate the influence of dual nutrient management strategies on the northern Gulf ecosystem. It is encouraging to see that the magnitude of nutrient reductions predicted by the model align with prior regression/statistical approaches that have formed the basis of nutrient management efforts in this system.

**Response:** We greatly appreciate the constructive and careful review.

With regard to the concept of ultimate versus proximate limiting nutrient: We find this a useful concept for framing our scientific questions and for putting the relative importance of N versus P in the broader context of nutrient reduction in aquatic systems. We feel that the concept provides a framework for clarifying some of the seemingly contradictory advice (e.g. Do we need P reductions for lakes, N reductions or dual nutrient reduction in coastal regions? What should be done if limitation switches between N and P?). In the Introduction there is quite a bit of text explaining the concept (see p. 2, line 8 to 25).

One of the two stated goals of our study is "to determine whether N or P is the ultimate limiting nutrient in this system, and to elucidate how their interplay affects hypoxia development" (p.4, line 1). We determine clearly that N is the ultimate limiting nutrient and that temporary P limitation has a very small effect on overall system productivity. To the best of our knowledge, this is the first time this has been clearly shown for the northern Gulf of Mexico or any other coastal system with dual nutrient limitation. We would like to add text to the Discussion to show more clearly how the concept applies and how our results confirm N as ultimately limiting nutrient, e.g., in subsection 3.2 "Sensitivity of PP and hypoxia to nutrient load reductions."

The Reviewer is correct that we address the concept only briefly in the Conclusions section. In the Introduction, we state that "establishing for a given estuarine or coastal system which of the two nutrients is the ultimate limiting one (on time scales of years to decades) should inform the design of sound nutrient-reduction strategies" (p.2, line 23). We would like to return to this statement explicitly in the Conclusions section.

**Specific Comments:**

1. Pg 1, line 23. The statement that coastal eutrophication from nutrient inputs is a growing problem. ... ignores decades of observations of the problem of eutrophication and nutrient

pollution. Perhaps it is now appropriate to state that coastal eutrophication from nutrient inputs is a long standing problem...

**Response:** Interesting point. We certainly didn't mean to ignore decades of work on eutrophication or imply it is a new problem. Instead we meant to emphasize that, even though eutrophication has been recognized as a problem decades ago, it is still growing (especially true for rapidly developing countries in Asia).

We propose to change the sentence as follows (new text in bold):
> "Coastal eutrophication as a result of anthropogenic nutrient inputs is a **long-standing and growing problem worldwide with negative effects…**"

2. Pg 2, line 13-14. N is not the main target of nutrient load reductions for Gulf hypoxia. In advance of the 2008 Hypoxia Action, EPA and the Hypoxia Task Force convened a special Hypoxia Advisory Panel thru the EPA SAB to review the science and provide recommendations for reducing Gulf hypoxia (see reference below). The SAB recommended a dual nutrient strategy of reducing N&P loads by 45%. The Gulf Hypoxia Task Force endorsed the dual nutrient reduction strategy and since the 2008 Action Plan a dual N&P load reduction has been and remains the target. The focus on N&P will have local in-stream water quality benefits as well as downstream water quality benefits in the Gulf consistent with our conceptual understanding and model predictions.

U.S.EPA 2007. Hypoxia in the Northern Gulf of Mexico. An update by the EPA Science Advisory Board. EPA-SAB-08-003. Washington D.C.

**Response:** This sentence is a general statement about nutrient reduction efforts targeting estuarine and coastal systems; those have generally focused on N. We will remove the reference to Task Force (2001), which is focused on the Gulf of Mexico and was included as an example only, to avoid the impression that we are talking about the Gulf here.

Further below (p. 3, line 5 to 14) we are talking specifically about management plans for the Gulf of Mexico. We had stated that nutrient reduction efforts "have long focused on N," but will modify this statement to "have **initially** focused on N." On line 12 we clearly stated that the Task Force has called for a dual nutrient strategy in 2008 and refer to the 2008 and 2013 Task Force publications. We will add the following text to refer to the U.S. EPA (2007) report:

> "In 2007, a special Hypoxia Advisory Panel was convened by the task Force and the U.S. Environmental Protection Agency and recommended adoption of a dual-nutrient strategy with the goal of reducing N&P loads *"by at least 45%"* (U.S. EPA 2007, p. ii)."

3. Pg 4, line 26-27. The monthly flux estimates described by Aulenbach et al., 2007 include two different regression model approaches; adjusted maximum likelihood estimates (AMLE) and the composite method. Please indicate which method was used.

**Response:** We would like to add text to indicate that the composite method is used.

4. Pg 5, line10. Please provide the rationale for choosing TN and DIP for load reduction scenarios rather than DIN and DIP, TN and TP or NOx and PO4, or something else. The literature is inconsistent in what is used for load reduction scenarios to predict Gulf hypoxic zone size. In the lower MS River, nitrate/nitrite comprises about 65% of the TN pool, and it is the nitrate/nitrite pool that has increased several-fold due to anthropogenic activities in the MS basin.

**Response:** We would like to add the following explanation:
"We chose to reduce TN because we assume the Task Force goals of reducing N load are referring to the sum on inorganic and organic N. It should be noted that a reduction in the organic matter load implies not only a reduction in N but also a slight reduction in organic P load. Conversely a reduction of organic P would imply a much larger reduction in N (by a factor of 16 if Redfield stoichiometry is assumed for the composition of organic matter). Hence we reduced only the inorganic P fraction in the DIP-reduction experiments."

5. Pg 10, line 3-6. The 30% N reduction goal described in the 2001 Hypoxia Action Plan comes from the work of V. Bierman and colleagues as part of the Topic 4 report for the first integrated assessment of hypoxia in the northern Gulf (see references below). Bierman et al used a version of the WASP model tailored to the Gulf hypoxia zone (which he called the NECOP model) to evaluate N and P reduction scenarios from 10% - 70%.

CENR (Committee on Environment and Natural Resources) 2000. Integrated assessment of hypoxia in the northern Gulf of Mexico. National Science and Technology Council Committee on Environment and Natural Resources, Washington, D.C., USA.

Brezonik, P.L., Bierman, V.J., Alexander, R., Anderson, J., Barko, J., Dortch, M., Hatch, L., Hitchcock, G.L., Keeney, D., Mulla, D., Smith, V., Walker, C., Whitledge, T., and Wiseman, W.J., 1999, Effects of reducing nutrient loads to surface waters within the Mississippi River Basin and Gulf of Mexico, Topic 4 Report for the Integrated Assessment on Hypoxia in the Gulf of Mexico: Silver Spring, Maryland, National Oceanic and Atmospheric Administration, NOAA Coastal Ocean Program Decision Analysis Series No. 18.

**Response:** Excellent, we appreciate the Reviewer pointing us to these sources. We would like to add the following text:
"The first estimate, based on box modeling work described in Brezonik et al. (1999) and published by the Hypoxia Task Force (2001), was a 30% reduction."

6. Pg 10, line 12. Suggest including reference to the statistical model of Turner et al. 2006 (see reference below), which helped to stimulate a series of additional statistical modeling approaches that you identify in the paper.

Turner, R. E., N. N. Rabalais, and D. Justic. 2006. Predicting summer hypoxia in the northern Gulf of Mexico: riverine N, P, and Si loading. Marine Pollution Bulletin 52:139–148.

**Response:** Yes, we would like to include this reference here.

7. Pg 10, lines 18-19. The statement that Greene et al 2009 (and Forrest et al 2011) predict >100% nutrient load to achieve the 5000 km2 goal is incorrect. Greene et al 2009 used 50% N and 45% N&P load reduction scenarios and found that model 11 (which used AMLE load estimates) could achieve the hypoxia target, whereas, model 12 (which used composite method load estimates) would not achieve the hypoxia tar get. These differences were attributed to differences in the AMLE and COMP load estimation methods, which influenced the model parameter coefficients. Thus, it would be correct to state on line 19 (and in Table 2) that >50% N reduction would be needed to achieve the hypoxia target using model 12.

**Response:** Apparently we misunderstood. We would like to remove the entries for model 12 and UEN from the table and the text.

8. Pg 17 Table 2. There is an inconsistency in abbreviations – figure caption state 'BO' refers to bioavailable N where the NCSU model in Scavia et al 2017 shows 'BN'. Finally, Greene et al 2009 used 'TP' not 'P' in their dual nutrient regression models.

**Response:** Yes, BO should be changed to BN. With regard to Greene et al., we would like to point out that, as indicated already in our response to comment 4, a reduction of organic P practically means a coincident reduction of organic N that is an order of magnitude larger than the reduction in P (because of the stoichiometry of organic matter). We are almost certain that Greene et al. did not apply such large N reductions in the regression analyses for P reductions, and hence would like to leave it at P rather than TP.

---

## Author Comment (AC2) · 3 Apr 2018

**Responses to Comments by Reviewer #2:**
https://doi.org/10.5194/bg-2017-470-RC2, 2018

Reviewer comments are pasted in their entirety in black font. Responses are in blue font.

This manuscript reports the first systematic analysis of the effects of single and dual nutrient load reductions from a spatially explicit physical-biogeochemical model for the northern Gulf of Mexico (GoM). Their manuscript is the next important step in their modeling efforts in the GoM. The manuscript tackles an important question regarding nutrient reduction strategies, it is well written and their work reports important simulations regarding dual nutrient reductions. They come to several additional important conclusions about the behavior of the GoM ecosystem including that reductions are more effective at reducing the area of hypoxia than in reducing primary production, and that the Gulf of Mexico system is saturated with nitrogen.

**Response:** We are grateful for the positive assessment and appreciate the thoughtful and constructive comments.

Some comments:

The US has not been able to make real nutrient reductions despite decades of voluntary controls. Since the GoM is N saturated, does that mean that it can't get worse as N loads continue to increase. Are we really at the maximum area given the current conditions and climate?

**Response:** Since the sensitivities are still well above zero, larger N loads would make the situation worse, just not as fast as they would at lower loads. We used the phrase "the system is approaching N saturation" and "already saturated in N" on page 8, lines 13 and 14. We would like to rephrase this to "is on a trajectory toward N saturation" and "almost saturated in N," because we don't want to imply that complete N saturation is reached.

Pg 1, line 8 "Evidence of P. . . since then" is awkward.

**Response:** Agree and would like to change to:
> "Since then evidence of P limitation during the time of hypoxia formation has arisen…"

Pg 3, line 11 One question regarding hypoxia is the legacy of a higher sediment respiratory demand following the build up of organic carbon stores in sediments with eutrophication (Turner et al. 2008) whereby repeated hypoxic events lead to an increased susceptibility of further hypoxia and accelerated eutrophication. I know one group in the GoM that believes that this process can not happen because of the physical conditions on the shelf would prevent the accumulation of organic matter stores. However, there are studies from the Baltic addressing the importance of the legacy of carbon and nutrients in the sediments. Could this be a factor in the GoM?

**Response:** Yes, it could be a factor in the Gulf, but whether it is remains an open question. Unfortunately our model can't address this question in its present form because we don't have an explicit sediment module that allows storage of organic matter in sediments. We would like to add the following text to address this:

> "Our model does not account for the possibility of a "legacy effect" as proposed by Turner et al. (2006, 2008). Turner and co-authors suggested that organic matter is accumulating in the sediments resulting in an increase in sediment oxygen consumption from year-to-year even as nutrient loads and system-wide productivity are stable. Our model does not include organic matter storage in sediments, and thus cannot address the question of legacy in its present form."

Pg 4, line 17 Revise to "as an additional"

**Response:** Agree

Pg 7, "Sensitivity" – this is an interesting concept. . .

Turner and Rabalais have examined the role of Si in influencing diatom growth and hence the sedimentation of organic matter. Have you tried any simulation with dissolved silica?

**Response:** Unfortunately we can't test the sensitivity to Si at present. The model does not include Si as a nutrient, only N and P. Including Si would be interesting, but would require significant changes in the model structure (i.e. including diatoms as a separate phytoplankton group, parameterizing the Si cycling through the zooplankton and detritus pools, and parameterizing the sediment remineralization of Si).

---

## Author Comment (AC3) · 3 Apr 2018

**Responses to Comments by Reviewer #3:**
https://doi.org/10.5194/bg-2017-470-RC3, 2018

Reviewer comments are pasted in their entirety in black font. Responses are in blue font.

1. This is an important and generally well-written modeling paper which moves the discussion of nutrient loading and Gulf hypoxia further along. In addressing the question of ultimate vs. proximate limiting nutrient, which I don't object to, probably the major point to make is that there would be no P limitation on the Louisiana shelf without the excessive loading of N from the Mississippi River, which the authors refer to as "saturating", an appropriate and significant term.

**Response:** We are grateful for the positive assessment and appreciate the careful review.

With regard to the comment that there would be no P limitation without the excessive N loading: We absolutely agree and should make this point clearly in the manuscript. In fact, our Figure 3 illustrates this point very well. We propose to add the following text in our revised manuscript:

> "The results in Figure 3 illustrate that reductions in P load would have a much smaller effect on system-wide primary production, than reductions in N load. In other words, P might be limiting temporarily, but that has little bearing on the overall system productivity. N is the ultimate limiting nutrient in this system. These results also imply that there would be little or no P limitation without the excessive N loads. As shown in Figure 3, at high N loads reductions in P load have a small effect on overall primary production, but this effect is much reduced for decreasing N loads and practically disappears for the 80% N load reduction."

2. Though the 2008 Hypoxia Task Force Action Plan, which is still the current goal though delayed in time, is mentioned on page 3 line 12, its load reduction goals (45% N and P) are not listed in Table 2 nor discussed in the text other than as a dual nutrient strategy, and the 2007 EPA Science Advisory Report, cited by another reviewer, which is the basis for these reduction goals, is not mentioned or cited. This SAB report was a 300-page major review of the science status at the time and the authors cite some of the papers important to this SAB review, but not the document itself.

**Response:** We are grateful that Reviewers 1 and 3 have pointed out this report and would like to include the reference in the Introduction (also see response to Reviewer 1) and in Table 2.

3. Unless I am reading it wrong, there is an inconsistency between the written legend to Fig. 2 and the labeled legend in the hypoxic area graph. The label says -60% P is light orange, and the written legend says that light orange is the -60% TN. The labeled legend in the figure appears to be correct and consistent with the following text, while the written legend appears incorrect. This labeling should be confirmed to be consistent throughout or it will prove very confusing.

**Response:** Indeed. We are glad the Reviewer noticed this. The figure legend is correct, but the caption is not and has to be corrected.

4. I agree that the use of TN and DIP river load reductions is confusing and probably has more of a historic rather than scientific origin. That said, much of the river Total P load is particulate

which then is solubilized when reaching the Gulf.

**Response:** As indicated in our response to Reviewer 1, we would like to add our rationale for this choice. We would like to add the following explanation:

> "We chose to reduce TN because we assume the Task Force goals of reducing N load are referring to the sum on inorganic and organic N. It should be noted that a reduction in the organic N load implies also a slight reduction in organic P. Conversely a reduction of organic P would imply a much larger reduction in N (by a factor of 16 if Redfield stoichiometry is assumed). Hence we reduced only the inorganic P fraction in the DIP-reduction experiments."